# Non-Conjugated Poly(Diphenylene Phthalide)—New Electroactive Material

**DOI:** 10.3390/polym15163366

**Published:** 2023-08-10

**Authors:** Danfis D. Karamov, Azat F. Galiev, Alexey A. Lachinov, Khalim I. Davlyatgareev, Sergey N. Salazkin, Artur R. Yakhin, Alexey N. Lachinov

**Affiliations:** 1Institute of Molecule and Crystal Physics—Subdivision of the Ufa Federal Research Centre of the Russian Academy of Sciences, 450075 Ufa, Russia; azat-red@yandex.ru (A.F.G.); lachinov_a@mail.ru (A.N.L.); 2Institute of Physics, Mathematics, Digital and Nanotechnologies, Akmulla Bashkir State Pedagogical University, 450000 Ufa, Russia; 3Nesmeyanov Institute of Organoelement Compounds of the Russian Academy of Sciences, 119334 Moscow, Russia; snsal@ineos.ac.ru; 4Skobeltsyn Institute of Nuclear Physics, Lomonosov Moscow State University, 119991 Moscow, Russia

**Keywords:** non-conjugated polymer, organic/organic interface, metal/organic interface, charge transport, organic electronics

## Abstract

In organic electronics, conjugated conductive polymers are most widely used. The scope of their application is currently very wide. Non-conjugated polymers are used much less in electronics and are usually used as insulation materials or materials for capacitors. However, the potential of non-conjugated polymers is much wider, due to the fact that new electronic materials with unique electronic properties can be created on the basis of non-conjugated polymers, as well as other inorganic dielectrics. This article demonstrates the possibilities of creating electrically conductive materials with unique electronic parameters based on non-conjugated polymers. The results of the study of the sensory properties of humidity are given as examples of the practical application of the structure. The abnormal electronic properties are realized along the interface of two polymer dielectrics with functional polar groups. The submicron films of polydiphenylenephthalide were used as a dielectric. It is shown that a quasi-two-dimensional electronic structure with abnormally large values of conductivity and mobility of charge carriers occurs along the interface. These structures are often called quasi-two-dimensional electron gas (Q2DEG). This article describes the manufacturing processes of multielectrode devices. Polymer films are deposited via the spin-coating method with polymer solutions in cyclohexanone. The metal electrodes were manufactured through thermal deposition in a vacuum. Three types of metal electrodes made of aluminum, copper and chromium were used. The influence of the electron work function of contacting metals on the electronic parameters of the structure was studied. It was established that the work function decrease leads to an increase in the conductivity and mobility of charge carriers. The charge carrier parameters were estimated based on the analysis of the current-voltage characteristics within the space-charge-limited current technique. The Richardson-Schottky thermionic emission model was used to evaluate values a potential barrier at metal/organic interfaces. It was established that the change in ambient humidity strongly affects the electronic transport properties along the polymer/polymer interface. It is demonstrated that the increase in conductivity with an increase in humidity occurs due to an increase in the mobility of charge carriers and a decrease in the height of the potential barrier at the three-dimensional metal contact with two-dimensional polymer interface. The potential barrier between the electrode and the bulk of the polymer film is significantly higher than between the electrode and the quasi-two-dimensional polymer structure.

## 1. Introduction

Surfaces and interfaces can have very different electronic properties to the bulk material [1]. The surface is the largest structural defect. The periodic potential is broken on the surface, and a high concentration of disrupted chemical bonds occurs [2,3,4,5]. As a consequence, an abnormally large concentration of surface electronic states is observed. The control of these states makes it possible to realize physical phenomena that can be embodied in various electronic devices. A more complicated situation arises when two surfaces interact. However, practically, the physics of all semiconductor devices is based on the realization of the properties of the interfaces of numerous combinations of different interfaces of metallic, semiconductor and dielectric materials.

The interface between two organic materials (OM) is widely used to achieve various practical purposes. The most well-known application is the tuning of electronic states in multilayer structures in order to optimize electronic (hole) transport in the metal/OM1/.../OMp/metal (semiconductor) heterostructure [6,7,8,9]. In addition, a nanometer thickness OM film embedded in this structure can localize the exciton recombination front. The latter technique is widely used in the creation of electroluminescent organic diodes [10,11]. Recently, along with traditional semiconductor materials, the use of wide-band materials has become relevant [12].

Interest in wide-band materials has increased due to the detection of metallic conductivity along the interface of two dielectrics of the perovskite class [13,14,15]. This is due to the appearance of a quasi-two-dimensional electron gas (Q2DEG) type along the interface of the electronic state. A similar phenomenon was detected along the interface of organic materials: tetrathiofulvalene/7,7,8,8-tetracyanoquinodimethane [16] and polydiphenylenephthalide/polydiphenylenephthalide [17]. These discoveries can be considered a fundamentally new technological approach for obtaining electronic materials of low dimension with high electronic characteristics. The active study of such quasi-two-dimensional structures is still ongoing [18,19,20,21,22,23,24,25].

Currently, conjugated polymers are most widely used in organic electronics [26,27,28,29,30]. Their molecular structure determines the delocalization of valence electrons. The scope of their application is currently very wide. Non-conjugated polymers are more often used as passive insulation materials or materials for capacitors [31,32,33,34]. However, it is obvious that the potential of non-conjugated polymers is much wider, especially if we take into account the possibility of creating quasi-two-dimensional electronic structures along the interfaces of these polymers.

Polymer heterostructures containing Q2DEG have a high conductivity compared to the bulk conductivity of individual materials. This conductivity is due to the abnormally high mobility of charge carriers. The occurrence of a Q2DEG state is associated with an effect similar to a “polarization catastrophe”, which is caused by a violation of the surface polarization field at the interface of two polar dielectrics. In [21], a correlation was established between the electronic parameters of the quasi-two-dimensional structure along the interface and the electronic parameters of the functional groups of the macromolecule. It can be seen that the best correlation between the electronic structure of organic molecules and the electronic properties of the interface with Q2DEG is described through surface polarization:(1)$$\Delta =\frac{4\pi }{\epsilon }nP_{0}\cos\left(\alpha \right),$$
where *n* is the surface dipole density, *P_0_* is the dipole moment of the functional group, α is the angle of the direction of the dipole moment relative to the surface and *ϵ* is the dielectric constant of the polymer. Thus, it is possible to change the electronic properties along the interface by choosing macromolecules with a suitable set (*n*, *P*_0_, α) [21]. Also, interface properties depend on the modification of the polymer/polymer interface, for example, by introducing a two-dimensional system comprising an island film of copper oxides [22].

Apparently, it is possible to observe the influence of the surface states of two-layer structures on the transport properties along the interface with a small thickness of polymer films forming the polymer/polymer interface. For example, this process is possible when a field of molecules is absorbed on the outer surface of the sample. Based on this principle, field-effect transistors are created and investigated, the role of the gate (or second gate) is provided through the potential of detectable polar molecules, particles and biological objects [34,35]. The high mobility of charge carriers can contribute to a change in conductivity along the interface when the field of adsorbed molecules changes. At the same time, improving the properties of the interface, the selection of materials and conditions for the implementation of high mobility along the interface are limited by the ohmic contact between the electrode and Q2DEG.

So, the purpose of the study is the investigation of the effect of adsorbed water vapor (located on the surface of the polymer/polymer bilayer structure) on the conductivity along the polymer/polymer interface.

## 2. Materials and Methods

### 2.1. Polydiphenylenephthalide

Polydiphenylenephthalide (PDPh) was chosen as the dielectric (Figure 1a). The synthesis of this polymer is described in detail in [36,37]. The temperature of the beginning of softening in air is about 420 °C, and the temperature of the beginning of decomposition ≥440 °C. This polymer is highly soluble in traditional organic solvents: cyclohexanone, chloroform, etc. PDPh is characterized by the following parameters: band gap ~4.2 eV, electron affinity ~2.0 eV, first ionization potential ~6.2 eV. These energy parameters were determined previously for the polymer by different research groups based on the analysis of experimental data [38,39] and quantum chemical numerical experiments [40,41]. PDPh has good film-forming properties at thicknesses of over 2 nm.

In addition, the choice of PDPh was due to the PDPh/PDPh boundary having a new type of collective electronic state, namely a Q2DEG. This was first discovered for non-conjugate polymers [17].

### 2.2. Samples

#### 2.2.1. Preparation of the Multilayer Structure

Previously, it was experimentally shown that a layer with a Q2DEG could be formed along the interface of PDPh films [17]. A two-dimensional area was formed at the interface of two polymer films by sequentially applying them to each other. Figure 1b shows a schematic representation of the manufactured structure. The layers were deposited by spin-coating from a polymer solution in an organic solvent. Glass slides (10 mm × 10 mm) were used as substrates for forming a structure with a polymer/polymer interface. The surfaces of the substrates were cleaned in an ultrasonic bath with organic solvents. The purification took place in the following sequence: acetone, ethanol, deionized water. The final cleaning was carried out with a desktop UV-ozone cleaner PSDP-UV4T (Novascan, Boone, NC, USA) at 100 °C. The extreme procedure also improves the wettability of the surface with a polymer solution. This has a beneficial effect on the uniformity of the thickness of the produced organic film.

The procedure for manufacturing an experimental structure consists of the following steps. The first stage is the lower polymer layer deposition. A ~300 nm-thick film was formed on a purified glass slides via spin-coating (Centrifuge ELMI CM-50, Riga, Latvia) at 3000 rpm from a 5 wt.% polymer solution in cyclohexanone for 1 min and was followed by solvent removal. The drying of the polymer film was conducted in two stages. Initially, the sample was dried in air for 1 h. This procedure is necessary for the slow removal of the main fraction of the solvent from the bulk of the polymer film. Then, the sample was dried in a drying chamber (AKTAN VTSh-K24-250, Fryazino, Russia) under pre-vacuum conditions at 150 °C for 1 h.

In the second stage, metal electrodes are formed on the surface of the first film. This is achieved through thermal evaporation in a vacuum of metal through a shadow mask on the polymer surface. The pressure of the residual gases in the vacuum chamber was 10^−6^ hPa. The gap between the electrodes was varied using a wire with a diameter of 10–60 μm. The thickness of the deposited metal layer was estimated using the mass of the evaporated metal and the distance from the evaporator to the sample surface [22]. A thickness of 150 nm was maintained for all samples. The selective control of the morphology of the surface of the metal film and the thickness of the layer was carried out via atomic force microscopy (AFM).

The third stage is the production of the top layer of polymer. The upper layer was made of the same polymer solution as the lower layer. All parameters of the technology of application and formation of the second polymer layer are similar to those of the first one.

#### 2.2.2. Selection of Metals for Electrodes

In a several papers [42,43], it has been experimentally shown that it is possible to form deep electronic states and a narrow conduction band near the Fermi level in thin dielectric films. In [42], this conclusion was based on the analysis of the current-voltage characteristics measured near the dielectric-conductor junction within the framework of the injection current model [44]. In [42], the energy distribution of electrons near the emission level (the Fermi level of the metal) was directly measured. The study of thermally stimulated phenomena [45,46] allowed us to obtain a detailed picture of the energy distribution of localized electronic states in the forbidden zone of the polymer material. Other methods also revealed the presence of deep traps [47,48].

In [49], it was found that the effective operation of the electrode output significantly affects the threshold characteristics of electronic switching in the metal/polymer/metal structure induced by a small uniaxial pressure. The bigger the effective work function (EWF) of the metal, the higher the threshold pressure.

Thus, the efficient operation of the electrode material work function can affect the conductivity along the interface of the two polymer films. In order to estimate the effect of the electrode material on the conductivity of the polymer/polymer interface, samples with various electrodes made of aluminum, copper and chromium (Sigma-Aldrich, St. Louis, MO, USA), with an EWF of 4.20, 4.36, 4.50–4.60 eV, respectively, were used [50,51].

#### 2.2.3. Characterization of Thin Films Using the AFM Method

The thickness and surface morphology control of polymer and metal layers was carried out on an NTEGRA II device (NT-MDT Spectrum Instruments, Zelenograd, Russia). A silicon probe with a typical force constant of 1.45–15.10 N∙m^−1^ and a guaranteed 10 nm radius of the tip curvature was used to scan the surface. AFM images were obtained using the semicontact method in the repulsion mode in atmospheric conditions at 23 °C room temperature. Image processing was carried out using the NT-MDT Nova Px software (NT-MDT Spectrum Instruments, Zelenograd, Russia).

The most difficult stage of creating a multilayer structure is the creation of an extended gap between planar electrodes. In this regard, special attention was paid to the control of manufactured electrodes on the polymer surface. Figure 2 shows the topography of the surface between metal electrodes made of copper and its 3D image. The gap between the electrodes was formed using a wire with a thickness of 15 μm. To align the image, the three-point plane alignment tool was used. The surface of the metal electrode was selected as the base surface. The light area represents the copper surface, the dark area represents the polymer surface. The transition between the surface of copper (light area) and polymer (dark area) is not sharp. The appearance of the intermediate layer is due to the large thickness and cylindrical shape of the wire, which prevents the maximum adsorption of metal on the polymer surface in the area of the gap between the electrodes. This form of mask allows the creation of a thinned electrode directly at the interface of 2 polymer films.

The average distance between the electrodes was determined statistically over the entire length of the gap and was about 10 μm.

### 2.3. Methods of Measurement and Analysis of the Results Obtained

#### 2.3.1. Current-Voltage Measurement

The current-voltage characteristics (IVC) of the quasi-two-dimensional area at the interface of two polymer films were carried out in ambient atmosphere, according to the scheme shown in Figure 3a. The control of conductivity is important in the interpretation of experimental data. For this purpose, before the stage of applying the first layer of polymer, a copper electrode was created on the surface of the substrate under the electrodes embedded in the interface at a distance of 2–3 mm from the gap (Figure 3b).

A Keysight B2902A precision source-measurement unit (Keysight Technologies, Inc., Santa Rose, CA, USA) was used to measure the voltage and current for the electrical characterization experimental sample. To assure a non-destructive measurement, the SMU was connected to the sample using a 4-wire connection on an MPI ETS50 manual probe station (MPI Corporation, Hsinchu, Taiwan) fitted with an MP40 micropositioner. This method allows for the elimination of contact and wire resistance.

#### 2.3.2. Humidity Measurements

The influence of humidity on the electrophysical parameters of the interface of polymer films was measured in a hermetical chamber with controlled humidity at room temperature 25 °C. The humidity level was changed by introducing water vapor into the chamber. To stabilize the humidity, the chamber was partially filled with silica gel. Humidity was measured with the ATT-5015 hygrometer probe (Aktakom, Novosibirsk, Russia).

## 3. Results

### 3.1. Polymer/Polymer Interface Properties

Figure 4 shows the current-voltage characteristics of six experimental samples of the same type with copper electrodes. The distance between the electrodes throughout the gap is about ~60 ± 5 μm. It is possible to observe the difference between the results of the IV measurements for the same types of samples. It was difficult to ensure the formation of identical boundaries between the electrode and the interface between the two polymer films during the manufacture of samples. Because during the formation of a polymer film via centrifugation, due to different centrifugal forces, there may be some difference in the orientation of the side fragments of molecules (angle α in Formula (1)). In addition, it was difficult to ensure the complete identity of the geometry of the shadow masks used in the deposition of metal electrodes on the surface of the polymer film.

The characteristics have a non-linear form. The sections of the linear dependence with a low potential difference and the regions of superlinear dependence are distinguished. The study of the electrophysical characteristics of experimental structures was carried out on the basis of the injection model of the space-charge-limited current (SCLC) [52,53]. This model has previously been successfully used to study the boundaries of organic dielectrics [17,21,22]. Figure 4b shows the IVC dependencies in semi-logarithmic coordinates. The type of dependence and the slope of the approximating line of all IVC in the range from 20 to 80 V is identical. The differences are most likely caused by the difference in the effective distance between the electrodes. Samples 1, 4 and 6 showed almost identical IVC, which are characterized by the highest conductivity. The further analysis and calculations of the electrophysical parameters were carried out according to the data of the IVC of these three samples.

The method of IVC analysis in the framework of SCLC is common for studying the mechanisms of charge carrier capture in insulators and semiconductors. It is known that the transport of charge carriers in organic materials is usually described by a model based on hops between localized states distributed in energy and space [54].

Within the classical framework SCLC theory described by Lampert [55], the capture of electrons into localized states in the band gap strongly affects the flow of current. The current conductivity is described by three limiting cases: Ohm’s law (I~U) at low voltages, Child’s law (or Mott-Henry’s law) for solids (I~U^2^) and a region with a limit filling of traps, which has a voltage threshold value and an extremely steep current increase. When interpreting experimental curves, the analysis comes down to finding the sections of the linear, quadratic or superlinear dependence of the current on the applied voltage I~kU^n^.

The analysis of the obtained results was carried out according to SCLC. The equilibrium concentration of charge carriers can be found through the equality conditions of the dependence equations of the current on the voltage for the linear Ohm’s section, i.e.,
(2)$$j_{1}=en_{0}\mu \frac{U}{L},$$
and a square plot. In the simple case when there are no traps and the SCLC is carried by carriers of the same sign, the current density is described by Child’s law:(3)$$j_{2}=\frac{9}{8}\;\varepsilon \varepsilon_{0}\mu \frac{U^{2}}{L^{3}},$$
where *j* is the current density, *L* is the distance between electrodes, ${U=U}_{n}$ is the voltage of the transition point of linear approximations of these sections in log–log coordinates, n_0_ is the equilibrium concentration of the charge carriers, μ is the effective charge carriers mobility, *e* is the elementary electronic charge, $\varepsilon_{0}$ = 8.85·10^−12^ F·m^−1^ is the electrical constant, and the permittivity of the material *ε* at room temperature is assumed to be equal to 2.7 [56]. Hence, equating Formulas (2) and (3), we obtain the expression for concentration:(4)$$n_{0}=\frac{9}{8}\frac{\varepsilon \varepsilon_{0}U_{n}^{}}{eL^{2}}.$$

Carrier mobility was calculated using the following equation:(5)$$\mu_{eff}=\frac{8}{9}\;\frac{jL^{3}}{\varepsilon \varepsilon_{0}U_{n}^{2}}.$$

The Richardson-Schottky thermionic emission (RS) model was used to estimate the relative change in the height of the potential barrier at the metal/polymer interface [57]. Assuming no image force lowering effect, a potential barrier can be defined using the following expression:(6)$$\varphi_{B}=\frac{kT}{e}\;ln\left(\frac{\;SA^{**}\;T^{2}}{I_{0}}\;\right),$$
where *T* is the temperature, *k* is the Boltzmann constant, *S* is the contact square, $A^{**}\;$is the Richardson constant and *I*_0_ is the saturation current. The experimental value of the saturation current is determined at the intersection of the direct linear approximation IVC in semi-logarithmic coordinates (ln(I) − U) with the U = 0 axis.

Thus, the quantitative analysis of the IVC makes it possible to estimate the transport parameters of charge carriers (effective mobility and concentration of their intrinsic charge carriers) and the potential barrier at the organic–metal interfaces.

To determine the electrophysical parameters of conductivity, an injection model and a formula for the Schottky barrier were used in accordance with Equations (4)–(6). The rearrangement of the IVC in logarithmic coordinates allowed us to establish that at low voltages a linear dependence is observed, which becomes superlinear when a certain value of $U_{n}$ is reached. It is known that the concentration of injected charge carriers reaches the concentration of their intrinsic carriers at this voltage. The values of the electrophysical parameters were estimated by averaging the data from 10 measurements on 5 samples of the same type with similar IVC.

The values of the concentration of charge carriers, the mobility of charge carriers and the height of the potential barrier were calculated. Its average value of conductivity is ~2 × 10^−10^ cm, the mobility of charge carriers is ~3 cm^2^/V∙s and the concentration of charge carriers is ~2 × 10^12^ cm^−3^. An estimation of the height of the potential barrier at the 3D_M_/2D_OM_ contact gave us a value of ~0.52 eV. The mobility of the charge carriers of the interface significantly exceeds the values characteristic of the thin polymer films of 10^−5^–10^−6^ cm^2^/V∙s, estimated by the time-of-flight method [58].

### 3.2. Influence of the Electrode Material

To assess the effect of various contact metals on the conductivity of the polymer/polymer interface, structures with electrodes made of metals with different work functions were made. Figure 5 shows the IVC of similar structures with electrodes made of aluminum, copper and chromium formed at the interface of polymer films. The distance between the electrodes was 45 μm, the average thickness of the electrode was 150 nm for all samples. The thickness of each polymer layer was 280 nm. The measurements were carried out in room conditions at a temperature of 25 °C and humidity of 30%.

The dependences of the current flowing along the polymer/polymer interface on the applied voltage of the samples have a nonlinear form. The analysis and evaluation of the charge carrier transfer parameters were carried out according to the above scheme and according to the Formulas (4)–(6). The obtained values of the calculated electronic parameters are presented in Table 1.

The concentration of proper charge carriers for all samples weakly changes in different electrode materials within the range of 10^11^ to 10^12^ cm^−3^. Such a weak dependence of the concentration on the electrode material is because the concentration of its intrinsic charge carriers is estimated according to the injection model. Since the same polymer material was used in the experiments, the concentrations should vary within the error of the experiment.

The estimates of the mobility of charge carriers and the values of electrical conductivity change when the work function of the contacting metal changes. Additionally, the maximum values are registered for the electrode with the minimum work function. At the same time, maximum conductivity is also observed. An increase in the work function leads to a decrease in mobility and conductivity along the polymer/polymer interface.

The work function change of the metal did not affect a significant change in the height of the potential barrier at the three-dimensional metal/quasi-two-dimensional structure contact. This fact may indicate the presence of the Fermi level pinning effect at the contact of the metal and the two-dimensional organic region. A similar phenomenon was previously observed, not only on conventional metal/polymer interfaces [6] but also on interfaces such as 3D metal/2D material [59]. In [60], it was experimentally shown that the pinning of the Fermi level at the metal–2D semiconductor interface can be caused by defects formed during the manufacture of electrodes. These electrodes were made using the method of high-vacuum electron beam evaporation of metal. The process of the thermal evaporation of metal in a vacuum is also a high-energy. In this regard, there are slight differences in the heights of potential Schottky barriers at the three-dimensional metal/quasi-two-dimensional polymer layer boundary (Table 1) depending on the work function of the electrode material. This may indicate the occurrence of defects at the contact area, which leads to the pinning of the Fermi level.

To assess the contribution of the bulk of the polymer film to the transport of charge carriers along the interface, an experiment was additionally performed to study the injection of charge carriers into the polymer film, as shown in Figure 3b. The assessment of the height of the potential Schottky barrier at the metal/polymer interface was carried out according to the scheme shown in Figure 3b. The heights Schottky barrier varied depending on the material of the injected electrode: for aluminum—0.79 eV; for copper—0.83 eV; and for chromium—0.85 eV. At the same time, the concentration of charge carriers was almost the same for all structures at ~10^15^ cm^−3^, and the mobility of their intrinsic charge carriers turned out to be extremely low at ~10^−10^ cm^2^/(V·s).

It is very important to understand how the change in the height of the potential barrier at the metal/polymer interface correlates with the change in the work function of the metal. To answer this question, the effective work function of the used metals was normalized to the effective work function of the Al. As a result, the following values were obtained: $\varphi_{Al}/\varphi_{Al}$ = 1; $\varphi_{Al}/\varphi_{Cu}$ = 0.96; and $\varphi_{Al}/\varphi_{Cr}$ = 0.93. A similar procedure was carried out with the values of the heights of potential barriers for metal/polymer interfaces. As a result, the following normalization values were obtained: $\varphi_{Al/PDPh}/\varphi_{Al/PDPh}$ = 1, $\varphi_{Al/PDPh}/\varphi_{Cu/PDPh}$ = 0.95; and $\varphi_{Al/PDPh}/\varphi_{Cu/PDPh}$ = 0.92. Taking into account the error in estimating the potential barrier using the Equation (6), the agreement between the results obtained is more than satisfactory. A change in the work function of the contacting metal causes corresponding changes in the height of the potential barrier at the metal/polymer interface.

On the one hand, this result can be considered as obvious. But on the other hand, we must remember that we are dealing with a metal/dielectric type contact. For this contact, it is generally assumed that the potential barrier at the interface should be determined by the difference between the work function of the metal and the energy of the electronic affinity of the polymer:(7)$$\varphi_{B}=\varphi_{M}-\xi_{P},$$
where $\varphi_{B}$ is the height of the potential barrier at the metal/polymer boundary, $\varphi_{M}\;$is the metal work function and $\xi_{P}$ is the electron affinity energy of the polymer.

However, previously [49], it had been shown that the height of the potential barrier is determined as the difference between the work function of metal and polymer in the case of polymer film materials:(8)$$\varphi_{B}=\varphi_{M}-\varphi_{P},$$
where $\varphi_{P}$ is the polymer work function. At the same time, the polymer work function is understood not as the energy equal to the ionization potential of the dielectric but as the energy of the Fermi level, defined as the upper level occupied by electrons with a probability of ½. For non-conjugated polymers, this level is often located in the middle of the gap between the upper occupied orbital and the lower vacant one. The experimental measurement of the height of the potential barrier confirmed the validity of this statement [61]. It should be noted that the inexplicably underestimated height of the potential barrier [62] in a structure with a polymer boundary had been repeatedly noted previously, corresponding to the value determined according to Equation (2). It is obvious that with close values of $\varphi_{M}$ and $\varphi_{P}$, the height of the potential barrier may be relatively small. In this regard, it can be asserted that at a constant value of $\varphi_{P}$, the transport of charge carriers across the polymer-metal interface will be determined by the work function of the electron from the injecting electrode.

Interphase charges in poly(vinylidene fluoride) (PVDF)-poly(methyl methacrylate) PMMA double-layered dielectric polymer thin films were investigated in [63]. The authors suggest that the polymer-polymer interface creates localized states of different depths, which lead to capture sites distributed over much greater energy levels. Since there are many localized states, the release or excitation of carriers in these states dominates the charge transfer process. These localized states act as carrier capture centers, and after capturing the injected charge from the electrodes, they become charged. This accumulation of spatial charge plays a key role in the transport of charge carriers perpendicular to the interface [64,65].

### 3.3. Influence of Humidity on the Conductivity and Electronic Parameters of an Experimental Structure

The electronic parameters of 2D electronic structures are sensitive to the molecules adsorbed on the surface of the structure. The field of this molecular layer can cause a change in the electronic spectrum of charge carriers due to the influence of an ordered layer of polar groups [66,67]. Apparently, a similar effect should be observed with the transport of charge carriers along the interface of two polymer films, because of the abnormally large mobility of charge carriers along the interface of two organic dielectrics.

Figure 6a shows the IVC along the polymer/polymer interface with embedded copper electrodes, depending on the humidity of the environment. Humidity in the chamber varied from 10 to 95% RH. The IVC measurements were carried out with a delay of 5 min after the establishment of the equilibrium relative humidity in the measuring chamber. IVC have a nonlinear appearance characteristic of the processes of a space-charge-limited current. An increase in humidity in the measuring chamber leads to a significant several-fold increase in conductivity. The analysis of the obtained IVC was carried out within the framework of the injection model, according to Formulas (4)–(6). The obtained results are presented graphically in Figure 7. As expected, an increase in conductivity with an increase in humidity is accompanied by an increase in the mobility of charge carriers (Figure 7a) and a decrease in the height of the potential barrier (Figure 7b). At the same time, the concentration of charge carriers decreases slightly.

However, the question is about a possible change in the conductivity of the thin polymer film with a change in humidity and, in this regard, the transport of charge carriers not along the interface but through the bulk of the polymer film. To answer this question, an additional electrode was used, which was pre-applied to the substrate under one of the electrodes embedded in the polymer/polymer interface (see Figure 3). Figure 6b shows the IVC measured at different ambient humidities. It can be seen that humidity has a weak effect on the measured IVC. Using these curves, the mobility of charge carriers, the height of the potential barrier and the concentration of charge carriers under the influence of humidity were estimated. These results are presented graphically in Figure 7a–c. No significant changes in these parameters from humidity were detected.

The initial values of the mobility parameters of the intrinsic charge carriers of the thin polymer film are more than 10 orders of magnitude less than the mobility of carriers along the interface. At the same time, the concentration of charge carriers in the bulk of the polymer film is four orders of magnitude higher. The concentration of charge carriers in the polymer film is primarily associated with the concentration of trap states. The higher the disorder, the greater the concentration of traps. The potential barriers at the interface of Cu/quasi-two-dimensional structure and Cu/polymer were 0.53 and 0.83 eV, respectively, at a minimum humidity of 5% RH.

Thus, it has been established that the contribution bulk to the polymer film mechanism of changes in conductivity along the polymer/polymer interface with changes in ambient humidity has not been detected.

### 3.4. The Effect of the Electrode Material Work Function on the Sensitivity of Electronic Parameters along the Polymer/Polymer Interface at Different Humidities

The effect of the electrode material work function on the IVC measured along the polymer/polymer interface at different ambient humidities is investigated. Figure 8a shows the dependences of the electrical conductivity of the experimental samples on humidity with electrodes made of copper, aluminum and chrome. The electrical conductivity was estimated from the IVC corresponding to the Ohm’s region of the current-voltage dependence, that is, for the area of intrinsic charge carriers. The nature of the dependence on humidity shows a nonlinear increase in conductivity for all samples. The results presented in Figure 8a show that the electrical conductivity along the polymer/polymer interface strongly depends on the electrode material. The estimates of the height of the potential barrier at the 3D/2D contact show a decrease in this parameter with an increase in the relative humidity of the environment (Figure 8b). The highest conductivity was registered for the electrode with the lowest work function operation, namely Al, and the lowest conductivity was registered for the electrode with the highest work function operation, namely Cr. In percentage terms, the decreases in the heights of the potential barrier at maximum humidity from the initial value are 7.5% in the case of an aluminum electrode, 6.5% in the case of copper and 5.5% in the case of chromium.

## 4. Conclusions

It has been established that a quasi-two-dimensional region is formed along the polydiphenylenephthalide/polydiphenylenephthalide interface, and it has an abnormally large conductivity for an organic dielectric. This region is characterized by high values of charge carrier mobility for organic materials. The maximum obtained value of mobility is 120 cm^2^/V∙s. The electronic properties of the quasi-two-dimensional structure depend on the properties of the contact of a three-dimensional metal electrode and a quasi-two-dimensional organic structure. In particular, it was found that a decrease in the effective work function of the electrode metal makes it possible to obtain a structure with bigger conductivity and mobility of charge carriers compared to metals with bigger work functions.

Changes in ambient humidity strongly affect electronic transport properties along the polymer/polymer interface. This increases the conductivity of the structure and the mobility of charge carriers. It is established that the bulk of the polymer film practically does not affect the transport of charge carriers when humidity changes. Perhaps this is due to the fact that the potential barrier between the electrode and the bulk of the three-dimensional metal/quasi-two-dimensional structure polymer film is significantly higher than the potential barrier between the electrode and the quasi-two-dimensional polymer structure.

## Figures and Tables

**Figure 1 polymers-15-03366-f001:**
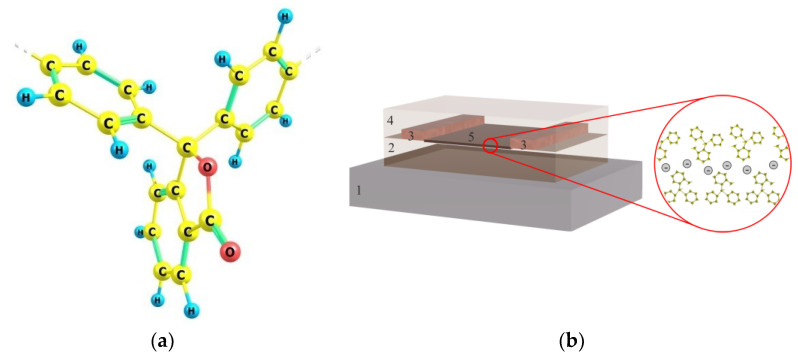
A structural formula of the monomer unit of PDPh (**a**) and a schematic representation of the multilayer structure (**b**). 1—substrates; 2—lower polymer layer; 3—metal electrodes; 4—upper polymer layer; 5—quasi-2D area at the between of two polymer films.

**Figure 2 polymers-15-03366-f002:**
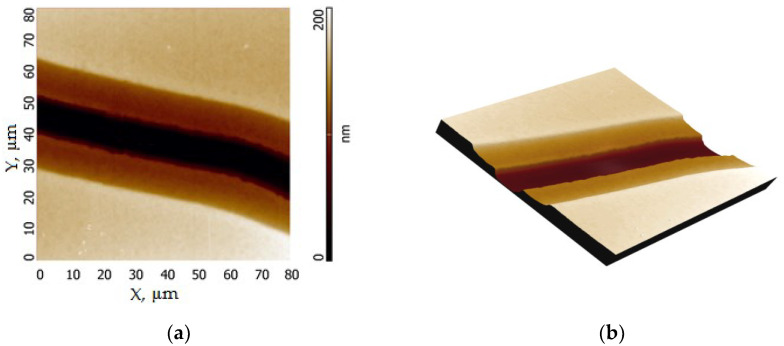
AFM topography of the experimental sample without the upper polymer film (**a**) and 3D image of the sample surface (**b**).

**Figure 3 polymers-15-03366-f003:**
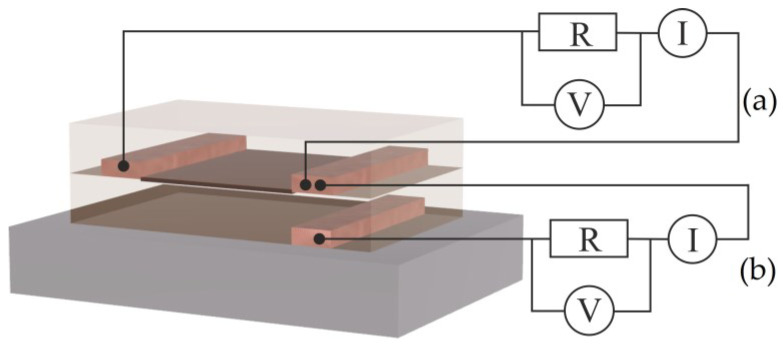
Schematic representation of the experimental sample and the scheme for measuring the IVC of the polymer/polymer interface (**a**) and the structure of the sample for measuring IVC at ambient humidity (**b**).

**Figure 4 polymers-15-03366-f004:**
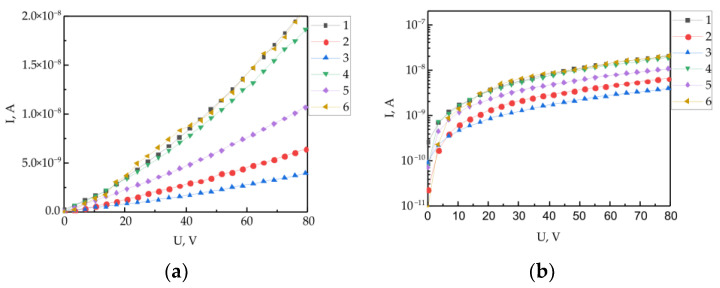
Current-voltage characteristics measured along the polymer/polymer interfaces of six samples of the same type (**a**) in semi-logarithmic coordinates (**b**).

**Figure 5 polymers-15-03366-f005:**
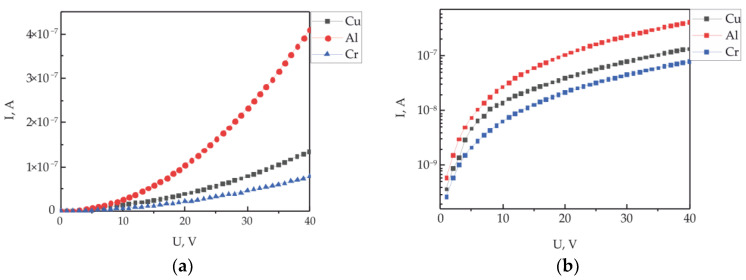
Current–voltage characteristics of the polymer/polymer interface depending on the electrode material (**a**) and in semi-logarithmic coordinates (**b**).

**Figure 6 polymers-15-03366-f006:**
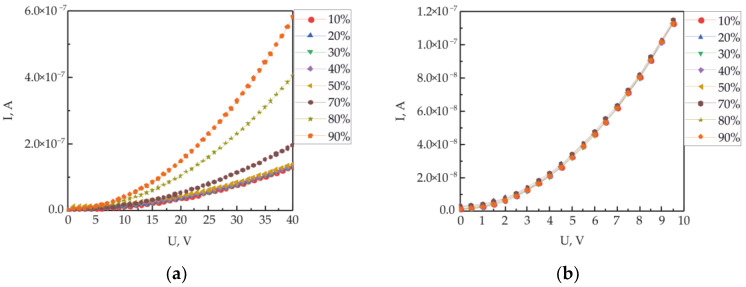
Current-voltage characteristics of the 3D_M_/2D_OM_/3D_M_ (**a**) and metal/polymer/metal (**b**) structure depending on the relative humidity of the atmosphere.

**Figure 7 polymers-15-03366-f007:**
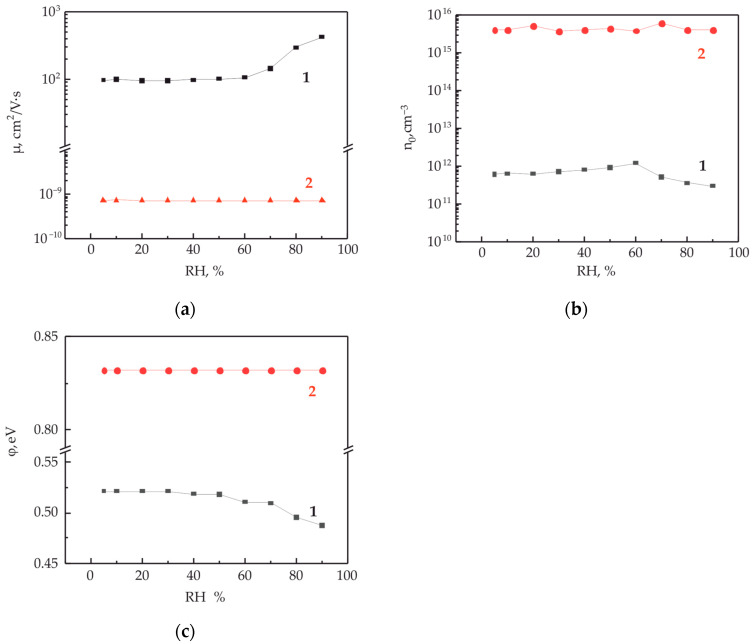
Charge carrier mobility (**a**) and concentration (**b**) of at the polymer/polymer interface (curve 1) and in the thin polymer film (curve 2), as well as a potential barrier at the metal/OM interface (**c**), depending on the relative humidity. The electrode material is copper.

**Figure 8 polymers-15-03366-f008:**
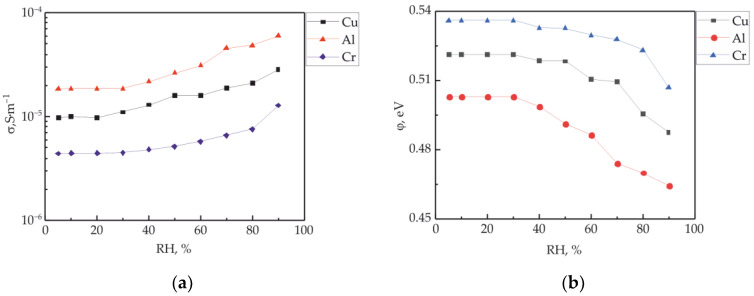
The specific conductivity of the polymer/polymer interface (**a**) and the change in the potential barrier of the three-dimensional metal/quasi-two-dimensional structure (**b**) depending on the humidity of the environment of electrodes made of different metals. The inset shows the notation of the curves.

**Table 1 polymers-15-03366-t001:** Electrophysical parameters of charge carriers and potential barrier at the three-dimensional metal/quasi-two-dimensional structure boundary, depending on the planar electrode material.

Electrode Material	Electrical Conductivity G, 10^−9^ S	Charge Carriers Concentration $n_{0}$, 10^11^ cm^−3^	Charge Carriers Mobility μ$,\;\frac{{cm}^{2}}{V\cdot s}$	Potential Barrier $\varphi_{B}$, eV	Metal Work Function [50,51] EWF, eV
Aluminum	10.3	1.56	120	0.50	4.20
Copper	3.5	7.26	95	0.52	4.36
Chrome	2.0	2.36	55	0.54	4.50

## Data Availability

Not applicable.

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
