# Peer review of "Non-Conjugated Poly(Diphenylene Phthalide)—New Electroactive Material"

_polymers, 2023, doi:10.3390/polym15163366_

Round 1

Reviewer 1 Report

The manuscript may be reconsidered after major revision as noted.

Influence of humidity on the conductivity is well known and there is nothing new from the authors

The HOMO and LUMO levels for the polymer should be calculated/presented from Cyclic Voltammetry.

Electrical conductivity in Copper is significantly lower (3 times) than Aluminium. Whereas, the metal work function is only 0.16 eV different. How? I suggest the authors to redo the control experiments of electrical conductivity measurements and present a details of new experimental results.

Also the authors must explain how a non-conjugated molecule can really alter the Schottky energy barrier at a molecular level. If it is not arising from the fundamental electronic properties, then what exactly is responsible?

I did not find any on/off ratio in presence and absence of humidity.

What is 1 and 2 should be mentioned near Figure 7.

Figure 7, metal used is not mentioned in caption, experiments needs to be cross-checked again as it is anomalous after 60% humidity.

Almost incomprehensible writing also unscientific at many points throughout the manuscript.

Almost incomprehensible writing also unscientific at many points throughout the manuscript.

The authors used incomplete sentence or expressions starting from abstract itself.

Improper use of adjective or adverb.

The phrase "It is established" has been used on 10 occasions and in some places very inappropriately as well.

Author Response

  1. The manuscript may be reconsidered after major revision as noted.

Influence of humidity on the conductivity is well known and there is nothing new from the authors.

- In this work, the influence is exerted by the change as a method of taking into account the features of 2D and 3D transport of charge carriers. A method for isolating the transfer of charge carriers in a quasi-two-dimensional size and in the volume of a polymer film. Also, the experiment is a model experiment for the study of sensory properties.

  1. The HOMO and LUMO levels for the polymer should be calculated/presented from Cyclic Voltammetry.

- Cyclic Voltammetry methods are effective for estimating the energies of HOMO and LUMO levels. We relied on the results of other equally useful measurements and calculations. The corresponding text was added to the article.

PDPh is a dielectric and is characterized by the following parameters in normal conditions: band gap ∼ 4.2 eV, electron affinity ∼ 2.0 eV, first ionization potential ∼ 6.2 eV. These energy parameters were determined earlier for the polymer by different research groups based on the analysis of experimental data [Zykov B.G. .et al. (1992). Valence electronic structure of phthalide-based polymers. Journal of electron spectroscopy and related phenomena, 61(1), 123-129.; Wu, C. R., et al. "Some chemical and electronic structures of the non-conjugated polymer poly (3, 3′-phthalidylidene-4, 4′-biphenylene)." Synthetic Metals 67.1-3 (1994): 125-128.], and quantum chemical numerical experiments. [Johansson, N., et al. (1994). A theoretical study of the chemical structure of the non-conjugated polymer poly (3, 3′-phthalidylidene-4, 4′-biphenylene). Synthetic Metals, 67(1-3), 319-322.; Yusupov, A. R., et al. (2019). Effect of Oxygen on the Conductive Properties of Thin Films of Nonconductive Polymer. Physics of the Solid State, 61, 450-455; Shishlov, N. M., & Khursan, S. L. (2015). Electron traps in poly (diphenylene phthalide) and poly (diphenylene sulfophthalide). Experimental manifestations and quantum chemical analysis. Russian Chemical Bulletin, 64, 766-790.].

  1. Electrical conductivity in Copper is significantly lower (3 times) than Aluminium. Whereas, the metal work function is only 0.16 eV different. How? I suggest the authors to redo the control experiments of electrical conductivity measurements and present a details of new experimental results.

- The reviewer is certainly right that the relationship between current and applied voltage depending on the configuration of the metal/polymer/metal structure is nontrivial. In the simplified case, the relationship between the current and the height of the potential barrier at the metal/semiconductor contact can be described by the Richardson-Dashman relation [Sze S. M., Li Y., Ng K. K. Physics of semiconductor devices. – John wiley & sons, 2021.]. According to this theory, the current depends on set of parameters of the structure and the external field. In particular, an increasing the potential barrier should lead to an exponential decreasing the flowing current. However, there are other parameters that are difficult to control in our experiment. For example, the effective mass of an electron. Changes in the electron effective mass can have a multidirectional effect on the magnitude of the flowing current. In this work, there was no task of estimating the effective mass of an electron depending on the configuration of a multilayer heterostructure and analyzing the effect of this parameter on the transport of charge carriers in the structure. As well as other parameters of Schottky theory. In our opinion, these fundamental issues can be the subject of a separate article. When interpreting the presented results, we draw the attention to the fact that other authors have previously observed similar features on other structures of the metal/polymer/metal type, and the proposed interpretations of these features do not contradict our results. So the results in Table 1 present the features of the transport of charge carriers in our structures.

Measurements of electronic properties on several hundred similar samples have shown that the presented results are in good agreement with the previously obtained results. Repeating the measurements, in our opinion, will not lead to significant correction of the results.

  1. Also the authors must explain how a non-conjugated molecule can really alter the Schottky energy barrier at a molecular level. If it is not arising from the fundamental electronic properties, then what exactly is responsible?

- The molecular mechanism of Schottky barrier formation at the metal/non-conjugated polymer interface may have been considered for the first time in [Duke C.B. and Fabish T.J. Charge-Induced Relaxation in Polymers. // Phys. Rev. Letters- 1976.- V. 37.- No.16.- P.1075 - 1078.]. In this work, it was found that an abnormally large decrease of the Schottky barrier can be a consequence of a complex mechanism of interaction of an excess charge injected from a metal electrode with a polymer molecule. Later, numerous experimental and theoretical results were obtained confirming these conclusions, for example, [I.Musa, W.Eccleston Electrical properties of polymer/Si heterojunctions //Thin Solid Films 343-344 (1999) 469-475.] [Arkhipov V.I., Emelianova E.V., Tak Y.H., and Bässler H.// J. Appl. Phys. 1998. V. 84. P. 848]. Similar studies are still ongoing.

The case of the 3D-metal / quasi-two-dimensional organic structure interface is also complicated. The work function change of the metal did not affect the height of the potential barrier at the 3D-metal / quasi-two-dimensional structure contact. This fact indicate the presence of the Fermi level pinning effect at the contact of the metal and the two-dimensional organic region. A similar phenomenon was previously observed not only on conventional metal/polymer interfaces [Fahlman, M. et al. Interfaces in organic electronics. Nat. Rev. Mater. 2019, 4(10), 627-650], but also on interfaces such as 3D-metal/2D material [Miao, J. et al. Recent Progress in Contact Engineering of Field-Effect Transistor Based on Two-Dimensional Materials. Nanomaterials 2022, 12(21), 3845]. In [Zhu, L. et al. The integrated spintronic functionalities of an individual high-spin state spin-crossover molecule between graphene nanoribbon electrodes. Nanotechnology 2015, 26(31), 315201.], it was experimentally shown that pinning of the Fermi level at the metal-2D semiconductor interface can be caused by defects formed during the electrodes deposition process. These electrodes were deposited by electron beam evaporation of metal. Process vacuum thermal evaporation metal is also a high-energy. In this regard, there is a slight difference in the height of potential Schottky barriers at the 3D-metal/quasi-two-dimensional polymer layer boundary.

  1. I did not find any on/off ratio in presence and absence of humidity.

- The study of the on/off ratio in the presence and absence of humidity was not the topic of the work.

  1. What is 1 and 2 should be mentioned near Figure 7. Figure 7, metal used is not mentioned in caption, experiments needs to be cross-checked again as it is anomalous after 60% humidity.

- Dependencies 1 and 2 and the electrode material have been indicated in the caption to the figures. The dependences obtained are typical for this polymer in the structure with copper electrodes. Measurements of electronic properties on several hundred similar samples have shown that the presented results are in good agreement with the previously obtained results. Repeating the measurements, in our opinion, will not lead to significant correction of the results.

  1. Almost incomprehensible writing also unscientific at many points throughout the manuscript. - We have improved the text according to the comments.

Reviewer 2 Report

-The manuscript “Non-conjugated polymers — new electroactive materials” focuses on the phenomenon of conductivity along the polymer/polymer interface. The discussion has a strong physical basis. In my opinion the title is too general. I noticed that this is a continuation of the Corresponding Author's work from more than 10 years ago Jetp Lett. 2010, 90, 726-730. The list of issues:

- The title is very broad while the research is very specific,  it focuses on quasi-two-dimensional electron gas based on polydiphenylenephthalide.

-  I am confused about the unit mkm. Is it unit from International System of Units?

-          Are the values in table 1 average? How many samples were measured under each condition?

Author Response

Thank you for your interest in our research results. 

  1. The title is very broad while the research is very specific,  it focuses on quasi-two-dimensional electron gas based on polydiphenylenephthalide.

- New title is «Non-conjugated Poly(Diphenylene Phthalide) — new electroactive material»

  1. I am confused about the unit mkm. Is it unit from International System of Units? 

- Corrected.

  1. Are the values in table 1 average? How many samples were measured under each condition? 

- The values in table 1 are averages. It was evaluated by averaging the data of 10 measurements on 5 samples of the same type produced in one technological cycle.

Sincerely, the team of authors.

Round 2

Reviewer 1 Report

I am satisfied with the revision. Therefore, the recommendation is accept.